# QTL Mapping for Resistance to Cankers Induced by *Pseudomonas syringae* pv. *actinidiae* (Psa) in a Tetraploid *Actinidia chinensis* Kiwifruit Population

**DOI:** 10.3390/pathogens9110967

**Published:** 2020-11-20

**Authors:** Jibran Tahir, Cyril Brendolise, Stephen Hoyte, Marielle Lucas, Susan Thomson, Kirsten Hoeata, Catherine McKenzie, Andrew Wotton, Keith Funnell, Ed Morgan, Duncan Hedderley, David Chagné, Peter M. Bourke, John McCallum, Susan E. Gardiner, Luis Gea

**Affiliations:** 1The New Zealand Institute for Plant and Food Research Limited, Private Bag 92-169, Auckland 1025, New Zealand; Jibran.Tahir@plantandfood.co.nz (J.T.); cyril.brendolise@plantandfood.co.nz (C.B.); 2The New Zealand Institute for Plant and Food Research Limited, Hamilton 3214, New Zealand; stephen.hoyte@plantandfood.co.nz; 3Breeding Department, Enza Zaden, 1602 DB Enkhuizen, The Netherlands; m.lucas@enzazaden.nl; 4The New Zealand Institute for Plant and Food Research Limited, Lincoln 7608, New Zealand; susan.thomson@plantandfood.co.nz; 5The New Zealand Institute for Plant and Food Research Limited, 412 No 1 Road, RD2, Te Puke 3182, New Zealand; Kirsten.hoeata@plantandfood.co.nz (K.H.); Catherine.mckenzie@plantandfood.co.nz (C.M.); 6The New Zealand Institute for Plant & Food Research Limited, Private Bag 11030, Manawatu Mail Centre, Palmerston North 4442, New Zealand; Andrew.Wotton@plantandfood.co.nz (A.W.); Keith.Funnell@plantandfood.co.nz (K.F.); Ed.Morgan@plantandfood.co.nz (E.M.); duncan.hedderley@plantandfood.co.nz (D.H.); David.Chagne@plantandfood.co.nz (D.C.); 7Plant Sciences Group, Department of Plant Sciences, Wageningen University and Research, Droevendaalsesteeg 1, P.O. Box 386, 6700 AJ Wageningen, The Netherlands; peter.bourke@wur.nl

**Keywords:** perennials, polyploid genetics, kiwifruit, polygenic resistance, bacterial pathogen, QTLs, chromosome pairing

## Abstract

Polyploidy is a key driver of significant evolutionary changes in plant species. The genus *Actinidia* (kiwifruit) exhibits multiple ploidy levels, which contribute to novel fruit traits, high yields and resistance to the canker-causing dieback disease incited by *Pseudomonas syringae* pv. *actinidiae* (Psa) biovar 3. However, the genetic mechanism for resistance to Psa observed in polyploid kiwifruit is not yet known. In this study we performed detailed genetic analysis of a tetraploid *Actinidia chinensis* var. *chinensis* population derived from a cross between a female parent that exhibits weak tolerance to Psa and a highly Psa-resistant male parent. We used the capture-sequencing approach across the whole kiwifruit genome and generated the first ultra-dense maps in a tetraploid kiwifruit population. We located quantitative trait loci (QTLs) for Psa resistance on these maps. Our approach to QTL mapping is based on the use of identity-by-descent trait mapping, which allowed us to relate the contribution of specific alleles from their respective homologues in the male and female parent, to the control of Psa resistance in the progeny. We identified genes in the diploid reference genome whose function is suggested to be involved in plant defense, which underly the QTLs, including receptor-like kinases. Our study is the first to cast light on the genetics of a polyploid kiwifruit and suggest a plausible mechanism for Psa resistance in this species.

## 1. Introduction

Most angiosperm species are diploid, the state where there are two complete sets of chromosomes in each nucleus. However, many plant species are polyploid, exhibiting more than two complete sets of chromosomes [1,2,3,4]. A number of agricultural and horticultural crops have their origin in polyploidy or are polyploid [5,6], for example banana, coffee, potato, cotton, strawberry and wheat. However, polyploids present a significant challenge to geneticists for the investigation of the genetic basis of traits, owing to the complex segregation patterns and preferential vs non-preferential pairing among chromosomes observed in such species [7,8,9]. Recent progress in sequencing and genotyping methodologies, as well as statistical tools for analyzing genetic data from polyploids, have overcome key issues that previously hindered the understanding of polyploid genomes [10,11].

The manner in which different homologous chromosomes pair and recombine during meiosis further determines the biological properties of a polyploid genome and the transmission of genetic information to the offspring. For example, in a tetraploid species where there is random pairing of homologous chromosomes, recombination occurs among all homologues. This manner of inheritance is termed polysomic and is characteristic of auto-polyploids such as potato and alfalfa, but can also be observed in “segmental” allopolyploids, such as peanuts [12,13,14]. Where the four homologous copies of a chromosome in a tetraploid species pair and recombine strictly preferentially, this results in disomic inheritance [13,14]. This pattern is typical of allopolyploids, such as wheat and coffee. In general, chromosomal pairing preference provides strong clues as to the origin of chromosomes in a species and its progenitors.

Pure auto- and allopolyploids are the two extremes of the spectrum and there are several species that fall in between, including some natural polyploid populations in the genus *Actinidia* (kiwifruit), which appears to be a “segmental” allopolyploid [15,16,17]. Additional polyploid kiwifruit selections have been generated by Plant & Food Research’s breeding programme by artificial chromosome doubling and crossing between different ploidy levels. Segmental allopolyploids can originate from parents belonging to distinct subspecies [18], resulting in the formation of templates for divergent genomes, with repercussions for chromosomal as well as transcriptional and epigenetic modifications [19,20,21]. This form of hybridity in genomes can contribute to homologous recombination among segments of chromosomes from respective parents [22], which entails changes at the molecular and phenotypic levels, paving the way for adaptation, speciation and invasiveness. In New Zealand, some of the native flora as well as introduced agricultural plant species exhibit allopolyploidy [23].

The exotic kiwifruit, with their high vitamin and mineral contents, attract consumers and contribute billions of dollars to the global economy, in addition to supporting the New Zealand horticultural industry. Originating in China, the *Actinidia* genus holds a resource of ploidy levels recognized at the sub-species level [24]. The green-fleshed hexaploid (2n = 6x = 174) *A. chinensis* var. *deliciosa* (‘Hayward’) was first cultivated in the mid-19th century, followed by the yellow-fleshed diploid (2n = 2x = 58) *A. chinensis* var. *chinensis* (‘Hort16A’) in the 1990s. ‘Hort16A’ succumbed to the global pandemic incited by the canker-causing *Pseudomonas syringae* pv. *actinidiae* (Psa) biovar 3 [25,26]. The genomes of diploid *A. chinensis* var. *chinensis* [27,28,29] and *A. eriantha* [30] have recently been sequenced, which will significantly assist genetic and genomic studies associated with agronomic traits, as well as elucidating the evolution of the genus and its fitness over diverse climates and geographies.

Breeding programmes focus on tetraploid cultivars, because of their resilience to the Psa disease, as well as robust fruit quality traits and size. However, there remains a substantial knowledge gap concerning the chromosomal biology and complex gene-trait associations in polyploid *Actinidia* compared with diploid species. *Actinidia* species are mostly dioecious and exhibit high heterozygosity [24]. This high degree of heterozygosity can facilitate trait mapping in segregating F1 populations using quantitative trait locus (QTL) mapping or association studies [11]. New genotyping platforms, including genotyping-by-sequencing (GBS) and single nucleotide polymorphism (SNP) arrays, are now available to capture allelic variations and haplotypes across the genome [31]. Furthermore, new tools have emerged for utilizing the abundant sequence data generated for the purpose of dissecting the genetic determinism of complex traits in polyploids, including the software TetraploidSNPMap [32] as well as packages based in R that include polymapR [33], net was [34], MapPoly [35], QTLpoly [36] and PERGOLA [37].

In this study, our goal was to cast light on the genetic regulation of resistance against the canker-causing Psa biovar 3 in a tetraploid F1 population of kiwifruit, by performing QTL mapping based on disease phenotype data collected in the field and greenhouse. A previous study has suggested that Psa resistance in a diploid yellow-fleshed kiwifruit population is governed by several QTLs [38]. However, the greater resilience against the pathogen, as well as low severity of disease symptoms compared with those in diploids [39], raises questions about the mode of inheritance of resistance in tetraploids. In the present study we utilized a robust new approach based on capture-based sequencing (Capture-Seq) to generate high density linkage maps using polymapR [33], followed by polyqtlR [40] for performing QTL mapping using identity-by-descent trait mapping [40]. Our analysis provides a first and comparative view of the genetic landscape of a tetraploid *A. chinensis,* and insights into the genomic regions regulating the quantitative form of Psa resistance in a yellow-fleshed sub-species of kiwifruit.

## 2. Results and Discussion

### 2.1. Bait Selection for High-Density SNP-Based Genotyping in Kiwifruit

A total of 9918 baits were designed by Rapid Genomics (Gainesville, Florida, USA) across the 29 chromosomes of diploid kiwifruit Red5, to genotype the population using Capture-Seq. These are exome baits designed within the genes to capture regions of 120 bp in length. Within each chromosome, 10–20 baits were spaced on average within every 1 Mb region. On average there were ~333 baits per chromosome (Appendix A).

### 2.2. Variant Calling and Dosage Estimation

A total of 9918 baits were used to genotype 235 F1 individuals of the tetraploid *A. chinensis* P1 × P2 population. These generated 1.03 × 10^9^ paired-end reads with a high quality score (mean Phred score 37). The population had a mean and median depth of 120x and 111x across baits, respectively. Within this dataset, a total of 725,175 raw SNP variants were identified that had an alignment rate of ~94.5% to the Red5 diploid kiwifruit genome. Estimates for self-relatedness in the population, calculated using the SNP dataset and read depth adjustments, indicated 20 individuals were outliers, suggesting that pollen contamination had occurred (Appendix A). After the removal of these contaminants, 13 F1 individuals with low quality (lower sample depth and call rate) were removed (Appendix A), which resulted in a final set of 188 true hybrids with high-quality sequence data for variant selection (Figure 1a).

A plot of the mean sequencing depth against the proportion of individuals with at least one sequence read at each SNP position (SNP call rate) indicates that most genotypes were called with sufficient read depth (1.00 log scale) (Figure 1b). A substantial proportion of SNPs had high minor allele frequency (MAF) (0.5), although SNPs with a lower MAF (0.275) were also common (Appendix A). Characterization of allele types by an allele frequency count in one of the seedlings demonstrated allele frequency rates in the dataset and was employed to represent the ploidy level of genotypes. Figure 1c,d highlight the frequency peaks of homozygous alleles/haplotypes (0/0 and 1/1) and heterozygous alleles/haplotypes (1/2, 3/4 and 1/4), respectively, for tetraploids. The 1/3 fractions in allelic frequencies in tetraploids have been previously noted and could point to possible null alleles (aaA0 or aAA0) [41] or other phenomena such as copy number variations in the repetitive portion of the reference genome [42]. Filtering of SNPs to a minimum read depth of 60 yielded ~70,000 markers usable for dosage estimation.

An empirical Bayesian analysis (Updog, “normal model”) to estimate genotype dosage in F1 individuals assists the visualization of the allelic dosage for SNP haplotypes. For bi-allelic SNPs, there are 5 possible dosage classes (ploidy +1) which are 0 (aaaa), 1(Aaaa), 2(AAaa), 3(AAAa) and 4(AAAA), where ‘a’ refers to the alternate allele while ‘A’ corresponds to the reference allele (Appendix A). Figure 1e comprises of 3 plots from Updog output, each demonstrating which genotype dosage is carried by the F1 individuals for a particular SNP. For example, each of plot 1 and 2 in Figure 1e demonstrates that the 188 F1 individuals are called to discrete classes as uniformly as possible for the respective SNPs, i.e., AAAA and AAAa in plot 1 and AAAA, AAAa and AAaa in plot 2. However, in plot 3 in Figure 1e we can observe that the dosage for the two major clusters of individuals is AAAA and AAAa, while a few individuals are called for genotype dosage 2, i.e., AAaa. This was not expected and could probably be due to systematic biases or over-dispersion for this particular SNP and hence qualifies for removal. Subsequent filtering of SNPs using Updog for a range of quality metrics, such as allele bias, over-dispersion and sequencing error rate, resulted in 60,006 usable bi-allelic markers representing 24 different marker types (Figure 2a).

### 2.3. Linkage Map Construction

From these 60,086 SNP markers, a subset of 45,436 were further selected after applying standard quality checks (including skewness, missing values at a threshold of 0.1 and duplication in individuals and markers) in polymapR. The 24 marker types (Figure 2a) were re-coded to nine simplified segregation types (1 × 0, 2 × 0, 0 × 1, 1 × 1, 2 × 1, 0 × 2, 1 × 2, 2 × 2, 1 × 3) (Figure 2b). For example, a marker which segregates as ‘0 × 1’ will segregate similarly to a marker with a segregation type of ‘3 × 4’. A figure depicting conversion of 4 possible simplex × nulliplex marker types (1 × 0, 1 × 4, 3 × 0 and 3 × 4) to a simpler segregation type (1 × 0) is presented in Appendix A. Initial construction of the expected 116 homologues (29 chromosomes × 4 homologues) was achieved using 19,563 homologue-specific simplex × nulliplex segregation types, including the re-coded marker types and consisted of 11,441 and 8122 markers for the P1 (1 × 0) and P2 (0 × 1) parents, respectively. A total of 4215 duplex × nulliplex (2 × 0, 0 × 2) and 4035 simplex × simplex (1 × 1) markers (Figure 2b) were then included to bridge the homologues of each linkage group (LG) from each parent. Finally, more complex allelic segregation types, such as simplex × triplex (1 × 3) (1514), simplex × duplex (1 × 2, 2 × 1) (4663) and duplex × duplex (2 × 2) (953) markers (Figure 2a) were added to improve the marker density of the maps and calculate the phase of each homologue.

The sex locus for kiwifruit is located on chromosome 25 [43,44], making this one of the chromosomes of particular interest (Figure 2c). We calculated the goodness-of-fit measure of the linkage maps by comparing multi-point estimates of pairwise recombination frequency from the genetic map, with the two-point estimates of recombination frequency. The weighted root mean square of the difference between these two estimates showed an inverse relationship with the logarithm of the odds (LOD) scores (Figure 2d). Where the difference between the expected and estimated recombination frequency was high, the LOD values were low (Figure 2d), indicating that the overall map quality was good. Figure 2e shows a plot of the position of markers on LG25 versus the estimate of recombination frequency with all other markers, demonstrating that tightly linked markers exhibit lower recombination frequency at the diagonal, depicted in green. Hence the estimated LOD score from the adjacent markers on the map would also be high, as represented by the red color in Figure 2f.

All 29 framework LG maps are depicted in Figure 2g, where each of the maps represents the integrated map derived from the phased maps of eight homologues, four from each parent. A total of 39,322 markers were assigned across the LGs, making this one of the highest-density linkage maps published for a polyploid species. The map lengths ranged from minimum 91 cM to maximum 143 cM, with a mean length of 115.5 cM. Genetic maps of LGs between 125 and 250 cM have been observed in various diploid, tetraploid and hexaploid species [45,46], with higher values often indicative of poor data quality or high map stress. Overall, the LG map lengths from this study indicate that a high-quality genetic map has been constructed.

To further assess the quality of our maps, we estimated the Identity-by-descent (IBD) probabilities [47,48] for nine seedlings in the population (Figure 3). Predicted allelotypes shown in dark colors have high probability of inheritance, while longer segments of parental haplotypes with relatively few light-colored bars represent regions of recombination breakpoints, which when taken together indicate a high quality dataset for QTL mapping.

### 2.4. Comparison of the Physical and Genetic Maps of Tetraploid A. chinensis and Quantification of Preferential Pairing

Marey plots enabled us to identify the degree of synteny between the tetraploid and diploid *A. chinensis* Red5 genome at the LG level (Figure 4). The baits used to genotype the tetraploid *A. chinensis* population carried positional information from the physical chromosomal scale assembly of the diploid kiwifruit genome Red5 [28], which was compared with the position of derived markers on the genetic maps.

We found that in most LGs, the marker positions as derived from the physical map of the diploid genome exhibited linear correspondence with the genetic positions in the tetraploid map. This indicates that, since polyploidization in *A. chinensis*, there have been only a few structural changes between the diploid and tetraploid genomes and that the order of the genes on the chromosomes of tetraploid *A. chinensis* may not be very different from that of the diploid *A. chinensis*.

However, the Marey plots suggest that for some chromosomes this situation does not apply for small segments of sequence, i.e., the marker genetic position does not follow the linear order of the physical map of Red5. These disruptions were visible on the plots for LG7, LG8, LG10, LG17, LG19, LG23 and LG27 and could arise from reads aligning to duplicated regions, from translocation events in the tetraploid genome, or possibly from artefacts from the mapping procedures (either in the physical assembly or the genetic map, or both). The Marey plots also depict regions on maps where the linear correspondence between the genetic and physical position of markers is weak, suggesting repressed recombination rates, for example the upper arm of LG25 (Figure 4), which corresponds to the sex-determining region and has been shown to have suppressed recombination rates in diploid *A. chinensis* [43,44]. More generally, these regions can be interpreted as putative sites of pericentromeric heterochromatin (i.e., the centromeres).

Finally, we tested the nature of pairing among chromosomes in tetraploid *A. chinensis* using our marker dataset. Our results show that pairing was random across most linkage groups, except for a few LGs with behavior deviating towards preferential pairing (at X^2^ < 0.001) (Figure 5). This included LGs 20, 23 and 27 in P1 and LGs7, 15, 22 and 27 in P2. LG27 exhibited preferential pairing in both parents, while LG3 in P1 has a strong but statistically weaker signature for preferential pairing. Hence, we find that the inheritance in tetraploid *A. chinensis* is mostly polysomic with preferential pairing in a few LGs. This is consistent with the cytological findings in tetraploid *A. chinensis* described earlier [16], and supports the view that tetraploid kiwifruit is a “segmental” allopolyploid.

### 2.5. Disease Phenotyping

#### 2.5.1. Field

The population was scored for resistance to Psa in two different orchards over the two years after planting. Best linear unbiased predictor (BLUP) values for Fld_Psa_scores were not Normally distributed (Figure 6a), with skewing towards the right of the order of −0.15 showing towards more disease. A non-Normal distribution was previously reported for the Fld_Psa_score in a diploid *A. chinensis* population [38]. Correlation of field observation breeding values between sites was high (r^2^ = 0.61), with between-site clonal repeatability of 0.75 indicating a strong genetic effect and a smaller effect for site in the variance component analysis.

#### 2.5.2. Stab Assay

Evaluation of five clonal replicates of each individual genotype in the population for Psa tolerance under greenhouse conditions inoculated using the stab assay revealed four key phenotypes characterizing the response to Psa infection. These included Psa-induced stem necrosis, ooze, wilt, and leaf spot, as well as a combined Psa score for each seedling in the population, calculations based on all four phenotypes. The distribution of Stab_Psa_score and Stem_necrosis across the population was Normal, whilst data for ooze, wilt and leaf spot were distributed in a non-Normal manner (Figure 6b–f). Normal distribution of the Stab_Psa_score was previously reported in a diploid *A. chinensis* population [38]. The analysis showed that greenhouse inoculation results are highly dependent on time of inoculation, similar to the earlier results for the diploid population [38], with clonal repeatabilities substantially lower than field repeatabilities. The best inoculated repeatability values were found for the sets inoculated in June (0.5) (winter) and the lowest for the sets inoculated in January (0.3) (summer).

Correlation between the greenhouse stab inoculation scores and combined field scores were poor (r^2^ = 0.04), indicating that the variables and responses measured in the greenhouse and field were different and that the ranking for the greenhouse inoculation set was a poor predictor of the field observations. Broad sense heritability values for greenhouse disease ratings calculated from seedling means were lower (0.0 to 0.51) and more variable than for the field observations. This finding was probably due to the large random effect found for inoculation time (owing to space limitations, higher temperatures in summer [49] and developmental differences, not all seedlings were inoculated at the same time of the year).

### 2.6. QTL Mapping

#### 2.6.1. Sex Locus

Because of their dioecy, the sex phenotype of kiwifruit plants plays an important role in commercial crop production. QTL mapping for the sex locus has not previously been performed in tetraploid kiwifruit and this provided us with the opportunity to validate tetraploid QTL mapping, using the flowering phenotype, before mapping for resistance. Flowering in kiwifruit over a whole population occurs in the third year of planting. However, 77 genotypes in the mapping population flowered in the second year, including 43 females and 28 males, enabling us to use this dataset to map the QTL associated with sex. We found a strong QTL on the upper arm of LG25 at 9.1 cM (LOD 17.2, PVE (percentage variance explained) = 64) over the marker Chr25:2290069 (Figure 7a). This region has previously been shown to contribute to the sex phenotype of the kiwifruit plants in diploid *A. chinensis* and the genes controlling sex-determination have been located on the Y-chromosome, LG25 [43,44]. The mapping of the phenotype to homologue 6, inherited from the male parent P2, can be clearly seen from the haplotype trace (as indicated by the high positive value associated with strong green color) in the IBD profile for LG25, below the QTL profile (Figure 7a).

The genotypic-information-coefficient (GIC) [47] was estimated across the population for each linkage group and used to assess the reliability of QTL estimation (Figure 7b). Where the GIC is close to one, estimates for unbiased QTL detection within that particular region are considered highly reliable. The GIC estimation for LG25 for both parents also demonstrates that GIC values can be variable at the telomeric region. This can affect the accurate positioning of a QTL in this area [47]; however, in the present case GIC is close to one at this position (~10 cM) on most homologues, including homologue 6. We can conclude that our trial QTL mapping for the sex phenotype in our tetraploid population identifies a male-specific genetic region on Y-chromosome that has been previously mapped in a diploid population and is used currently to predict sex for pre-breeding, clearly demonstrating that QTL mapping has been effective in our population.

Our approach for studying the genetic basis of qualitative and quantitative traits in a tetraploid kiwifruit population was based on recent advances in QTL mapping in plant species that exhibit higher ploidy, and achieved using high-throughput genotyping with Capture-Seq, as well as access to new tools for genetic and QTL mapping for complex genomes of plant species [10,46,47,50,51].

#### 2.6.2. Psa Resistance

QTL mapping of Fld_Psa_score, using data from two years of phenotyping, revealed four QTLs for resistance to Psa in the field. The first QTL is located on LG1 at 68.1 cM (LOD 4.5, PVE = 11.3), over the marker Chr1: 14996600 (Figure 8a). The 1-LOD support interval for this QTL lies between 58 and 72 cM. The second QTL is located on LG2, with its peak at 93.3 cM (LOD 5.0, PVE = 9.7) over marker Chr2:13522420 and a 1-LOD support interval between 89 and 99.1 cM (Figure 8b). The third QTL is located on LG4 at 54.4 cM (LOD 4.5, PVE = 11.2) over the marker Chr4:5917168, with the 1-LOD support interval for this QTL located between 45 and 63 cM (Figure 8c). The final QTL peak is detected on LG7, at 59.2 cM (LOD 5.1, PVE = 12.8) over underlying marker Chr7:6815185. The estimated 1-LOD interval support for this QTL peak is between 53 and 63 cM. These QTLs were significant at *p* < 0.05 and *p* <  0.1 using a chromosome-wide permutation test (Figure 8d). The contribution of homologues under each of the QTLs is indicative of the source of Psa resistance in the population. Taken together, P2, which is the more resistant parent, has three QTLs, including one at h8 under LG1, and two at each of the h5 of LG4 and LG7 which contributes towards field resistance or quantitative resistance (QR) against Psa, while P1, which is weakly tolerant, also contributes a minor QTL for Psa resistance on h2 of LG2. Previously it was shown that in diploid kiwifruit, a QTL for resistance to Psa is located in a Psa susceptible cultivar ‘Hort16A’ [38]. Similarly, other studies have pointed out towards alleles from susceptible parents involved in delivering varying degree of resistance against different pathogens [52,53,54].

To determine if the QTLs act in combination in delivering QR we next calculated the PVE combined, to assess the additive effect of all four QTLs in combination. The 4-QTL model (Q1(LG1)_Q2(LG2)_Q3(LG4)_Q4(LG7)) indicated a total PVE of 34. We suggest that the four QTLs act in combination in the progeny to provide a more robust QR against Psa in the field. Notably, these genetic regions do not overlap with those identified in previous work on locating QTLs controlling QR against Psa in the yellow-flesh diploid *A. chinensis* [38], suggesting that the origin of Psa resistance in tetraploid species is not related to any common ancestral loci in *Actinidia* species and that the sources of QR might be diverse in terms of their positions in the kiwifruit genome in between the sub-species.

Our results for QTL mapping for control of resistance to Psa using data from the field suggest that Psa resistance in the tetraploid *A. chinensis* population is a polygenic trait which is likely to be controlled by multiple genes. This is consistent with a previous hypothesis for mode of Psa resistance in tetraploid kiwifruit populations [39]. We further demonstrated which key homologues from P2, as well as P1, contribute to these QTLs that act in combination to exhibit Psa resistance in the field. Further validation of the QTLs from the Field_Psa_score needs to be performed in backcrosses as well as out-crosses in tetraploid populations which share either P2 in their pedigree or grandparents in the F1 population.

QTL mapping of the response to Psa inoculation measured in the stab assay identified two major QTLs for the Stab_Psa_score on a pseudo-autosomal region on LG25 (LOD 6.9 at 107.1 cM with underlying marker Chr25:19357486) and LG29 (LOD 5.9 at 73.4 cM over marker Chr29:11615148) (Appendix A). One QTL was identified for Stem_necrosis on LG21 (LOD 5.2, at 37.8 cM over marker Chr21:3833187), as well as a single QTL for Leafspot on LG16 (LOD 4.3 at 34.8 cM over marker Chr16:3574359) (Appendix A). Notably, no significant QTLs were detected for the Ooze and Wilt phenotypes. The QTLs for phenotypic data from the Stab assay did not overlap with the QTLs from the Fld_Psa_Score. We found a similar scenario in our previous study, where QTLs from the stab assay did not overlap with the QTLs for Psa resistance in diploid *A. chinensis* [38]. It seems plausible that the expression of Psa resistance is influenced by the mode of infection in the stab assay, compared with the field phenotyping, as well as being intricately dependent upon developmental stage, environment, and tissues.

### 2.7. Candidate Genes for Control of Field Psa Tolerance Underlying the QTLs

The genes determining the QTLs identified using the Fld_Psa_score phenotype are expected to play an important role in determining the genetic basis of control of Psa resistance in tetraploid *A. chinensis*. Our search among the gene models in the diploid Red5 genome, underlying the genetic markers flanking the four major QTLs yielded genes related to signaling, transporter, and various other functions related to carbohydrate metabolism, transcription and stress tolerance. Appendix A lists all the predicted gene models identified on the diploid Red5 genome at the locations underlying the four QTLs. From these we have highlighted a few, most of which are associated with plant immunity, including Acc00939.1 (*Proline-rich receptor-like protein kinase PERK9-like*), Acc00941.1 and Acc00956.1 (*Receptor-like serine/threonine-protein kinase ALE2*) as well as Acc00949.1 (*Wall-associated receptor kinase-like*) at LG1; and Acc04873.1 (*WD and tetratricopeptide repeats-like*), and Acc04875.1 (*Serine/threonine-protein kinase-like*) Acc04916.1 (Sugar transport protein 5) on LG4. Five genes are annotated on LG2 with putative functions associated with plant immunity to pathogens. These include Acc02420.1 (*Glycerol-3-phosphate acyltransferase-like*), Acc02421.1 (a *Protease Do-like 1, chloroplastic-like*), Acc02426.1 (*Serine/threonine-protein kinase-like*), Acc02438.1 (*High affinity nitrate transporter-like*) and Acc02443.1 (*Mitogen-activated protein kinase 3-like*). A further five genes involved in plant-pathogen interaction were identified under the QTL on LG7, including Acc07849.1 (*Transmembrane protein adipocyte-associated 1-like*), Acc07852.1, Acc07869.1 and Acc07861.1 (*Serine/threonine-protein kinase EDR1-like*) and Acc07858.1 (*WD repeat-containing protein 48-like*).

Overall, our results indicate that most genes underlying QTLs for control of resistance to Psa in tetraploid *A. chinensis* belong to a gene network that encodes proteins which have putative functions in membrane and intracellular signaling, such as receptor-like (RL) serine/threonine kinases and anion transporters, as well as putative enzyme functions. There seems to be functional commonality in the genes that underlie the QTLs identified for Psa resistance in the field in both diploid [38] and tetraploid *A. chinensis*; for example, the gene family that encodes RL kinases. RL kinases are known to play a significant role in plant immunity, especially early pathogen recognition and activation of plant defense [55,56,57,58]. The mechanistic basis of resistance for the two sub-species may rely on combinatorial action of defense proteins providing strong surveillance for pathogen-associated molecular patterns (PAMPs) and activation of signaling cascade for PAMPS-triggered immunity [59]. Our findings pinpoint gene families, genetic regions and mechanisms underlying resistance to Psa in diploid [38] and tetraploid yellow-fleshed *Actinidia* species.

## 3. Materials and Methods

### 3.1. Plant Material

The F1 population for genetic mapping of resistance to Psa was developed from a cross between a moderately Psa-tolerant female ‘P1’ and a highly resistant male ‘P2’. The designations and a pedigree are provided for these parents in Appendix A. A total of 235 F1 genotypes were germinated aseptically in 2017, under standard tissue culture growth conditions. Each seedling was replicated 15 times by cuttings, either using tissue culture, or under standard greenhouse conditions, prior to phenotyping in the field, or with stab bioassays. A set of 235 seedlings with five clonal replicates each was planted in the Plant & Food Research orchard in Te Puke (Bay of Plenty, New Zealand, −37° S and 176° E) and a similarly replicated set of the same seedling was planted in the Plant & Food Research orchard in Kerikeri (Bay of Islands, New Zealand, −35° S and 173° E) in a randomized block design in 2018. Both locations, Te Puke and Kerikeri, are located within the main kiwifruit-growing regions of New Zealand and are affected by natural infections of Psa (Kiwifruit Vine Health: https://www.kvh.org.nz/maps_stats). The mapping population was maintained under standard orchard conditions, without any control for Psa infection, in both research orchards for two years. All the seedlings which were replicated for the field trial were also replicated five times for the stab assay. Plants were grown up to 1 m in height in pots in the greenhouse and then subjected to the stab assay in batches, as described previously [38,60].

### 3.2. Phenotyping

The field population was assessed for symptoms in response to natural Psa infection between 2018 and 2020. Traits scored included cane death, ooze, shoot death and tip death. A cumulative Psa score (Psa_score_Field) was calculated after two years, as described previously [38]. Least Square Means (LSMs) were calculated for this score for each genotype and BLUPs were estimated.

The stab bioassays were performed as described previously [38,60], in standard greenhouse conditions between September to December (spring) and February to April (summer) for two years under standard greenhouse facilities at Plant & Food Research Ruakura, Hamilton. Inoculations were performed with the Strain 10627of Psa biovar 3 [61], in the greenhouse at temperatures ranging between 22 and 30 °C. The flowering data for the population were obtained in 2019.

### 3.3. Genotyping, Genetic Map Construction and QTL Mapping

DNA was extracted from freeze-dried leaves using the Cetyl trimethylammonium bromide (CTAB) method [62] and ~1 µg of DNA was provided to Rapid Genomics LLC, Gainesville, Florida for Capture-sequencing. The DNA library preparation and capture enrichment were done by Rapid Genomics LLC). They designed approximately 10,000 baits in the Red5 genome [28] to capture genetic variations, targeting putative SNPs, using high throughput Illumina^®^ sequencing, which generated 150-bp paired-end reads. Raw reads were assessed for quality using FASTQC (https://github.com/s-andrews/FastQC) and trimmed using Trimmomatic [63]. Reads were aligned to the Red5 genome using Bwa-MEM [64] and Samtools [65].

On-target statistics were generated using TargQC (https://pypi.org/project/targqc/): mean read counts 3.75 × 10^6^, mean mapping rate of 98% and a padded (i.e., ±200 base padding) on-target mean of 68%. Target coverage depth was estimated using bedtools; the mean across all baits for each individual was estimated. Variant calling and genotyping were performed using Freebayes-1.1.0 (command “ -p 4 -C 5 -k --report-genotype-likelihood-max --min-mapping-quality 30 --min-base-quality 20 ” as described [66].

SNP calling was undertaken against the *A. chinensis* Red5 genome (1.69.0) [28]. The Variant Call File file for SNP dataset was analyzed using vcfR [67] to extract total, alternate and reference allele counts with a minimum read depth of 60, to filter out low-quality SNPs. Kinship relatedness, Minor Allele Frequency (MAF) and Principal Components Analysis (PCA) plots were calculated and constructed using the KGD software [68]. Dosage estimates were performed using an empirical Bayesian approach for dosage genotyping in offspring (Updog software, “normal model”) (command “ ## Remove SNPs with mean read-depth below some value: meandepth <- rowMeans(DP, na.rm = TRUE), mindepth <- 60, mean(meandepth < mindepth), DP <- DP[meandepth >= mindepth, ], RO <- RO[meandepth >= mindepth, ], ## Fit updog using reference and total matrices: parallel::detectCores() ## max number of cores to use, mout <- multidog(refmat = RO, sizemat = DP, ploidy = 4, model = "norm", nc = 6) ”) [69]. SNPs were further filtered for quality, allele bias, over-dispersion and sequencing errors using UpDog software (command “filter_snp (mout, bias > 0.5 & bias < 2 & seq < 0.01 & od < 0.02)”).

Linkage mapping analysis and phase calculation were performed using polymapR [33] (1.1.0), which further filtered the dataset for high quality SNPs (qall_mult > 0 using function checkF1). Linkage maps were built chromosome by chromosome, owing to the high number of genetic markers and linkage groups and the amount of computing required. Once generated, the individual Linkage Group (LG) phased maps were merged for QTL mapping using polyqtlR [40]. This software performs QTL mapping in the F1 populations of outcrossing, heterozygous polyploid species and relies on identity-by-descent (IBD) probabilities to perform interval mapping, following the approach proposed by Kempthorne [70] and later developed by Hackett [71,72]. The software was made available to this project by Peter Bourke and will soon be available through the Comprehensive R Archive Network (CRAN: https://cran.r-project.org/). Preferential pairing among LGs was calculated using the IBD estimation function polyOrigin, which is part of the polyqtlR package, using the default function test_prefpairing. Percentage variance explained by each QTL and the combined model was calculated using the PVE function in polyqtlR. Owing to the large number of LGs and high density of markers, LOD thresholds were calculated for each LG separately, using a permutation test (1000 permutations, at α <0.01 and <0.05). Genes underlying the QTLs were identified from the in-house kiwifruit gene model database for the diploid Red5 genome version 1.69.0 [28]. The positions for the genomic regions were derived from the location information present within each marker.

## 4. Conclusions

Our study has demonstrated the application of capture-sequencing to generate a high density SNP-based linkage map in a heterozygous polyploid fruit crop. This is the first study to report the genetic maps of a polyploid *Actinidia* species. The data showed that the tetraploid kiwifruit *A. chinensis* exhibits polysomic inheritance for most LGs, as well as disomic inheritance for several LGs, fulfilling the characteristics of a ‘segmental’ allopolyploid. IBD-based QTL mapping for control of the response to Psa infection in two different field sites enabled us to map four QTLs for control of Psa resistance in the field in this population, identify the alleles on parental homologues and the additive effect exhibited in the variance. The candidate genes underlying the QTLs will further be explored to develop an understanding of their roles in strong quantitative basal resistance in kiwifruit, and this may assist in the development of high-throughput genetic markers. Our study will help advance the development of marker assisted selection of tetraploid yellow-fleshed Psa-resistant kiwifruit seedlings, following validation of the QTLs. The findings from the stab assay further inform us that the quantitative nature of this trait is affected by a complex of genetic, environmental and developmental factors, as well as the method of inoculation and the phenotypes scored for QTL mapping.

## Figures and Tables

**Figure 1 pathogens-09-00967-f001:**
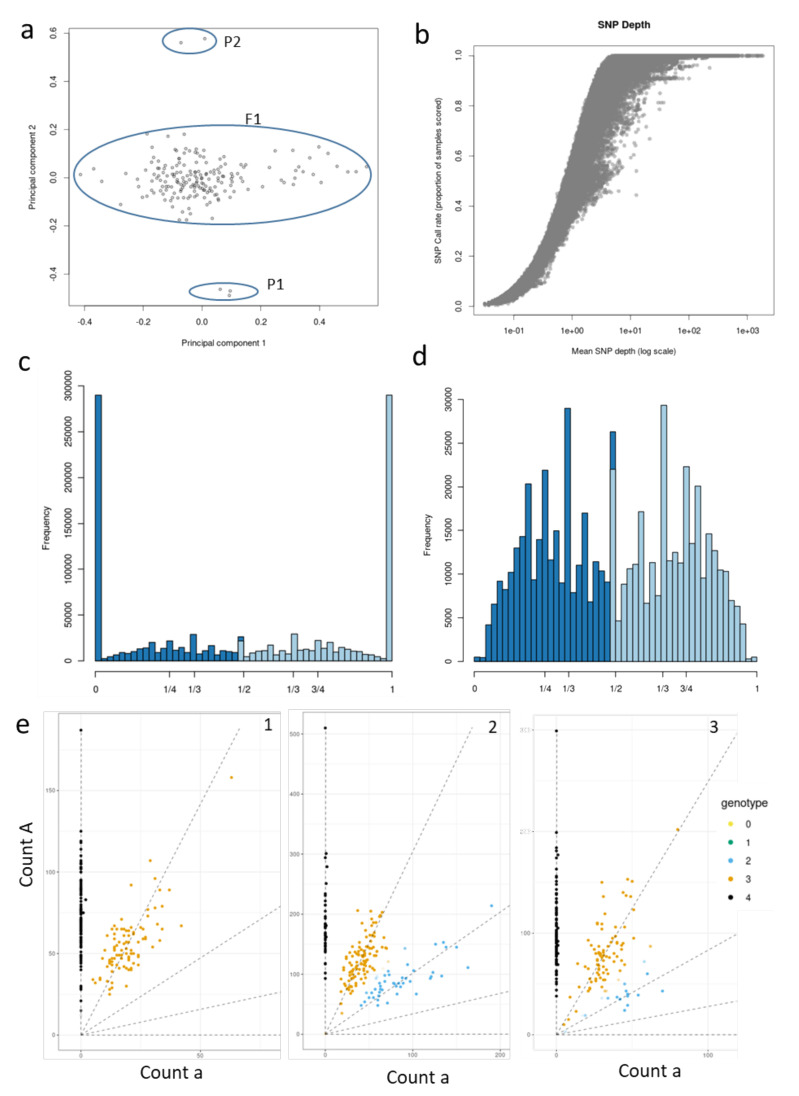
Variant calling and dosage estimates in a tetraploid *A. chinensis* F1 population. (**a**) Principal component analysis was performed by KGD software on the estimates of self-relatedness using the genetic dataset and the read depth information of the population (F1) and the parents P1 & P2; (**b**) is a plot of SNP call rates (which is the proportion of individuals observed with at least one allele) versus mean SNP depth (log scale); (**c**) shows allele frequency in one of the F1 genotypes for the most common class of homozygous variants (0 and 1); (**d**) shows heterozygous variants in the same genotype as in (**c**), with peaks at 1/3 and 1/2. Allele frequencies are 0, 1/4, 1/3, 1/2, 1/3, 3/4 and 1; (**e**) shows three different plots for genotype dosage estimates in F1 individuals for three different bi-allelic SNPs (1–3). The dots represent individuals between the read counts for the reference allele (A, y-axis) and the counts of reads for the alternative allele (a, x-axis). The dots are colored based on the five genotype dosages in tetraploids (ploidy+1) such that the genotypes 0, 1, 2, 3 and 4 refer to dosages aaaa, Aaaa, AAaa, AAAa and AAAA respectively.

**Figure 2 pathogens-09-00967-f002:**
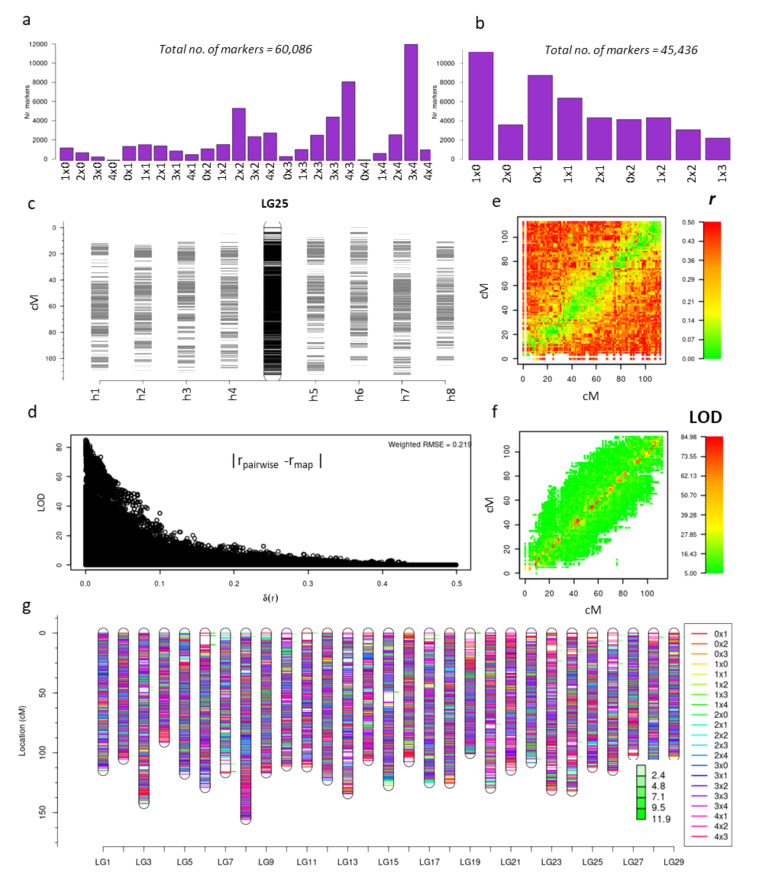
Linkage map construction in a tetraploid *A. chinensis* F1 population. (**a**) is the plot of the number of markers in each of 24 Single Nucleotide Polymorphism (SNP) marker classes in the population; (**b**) number of SNP markers after re-coding marker types and filtering for quality; (**c**) shows the fully phased and integrated map of Linkage Group (LG)25 (center), with markers on each homologue of P1 (h1–h4) and P2 (h5–h8) to left and right, respectively; (**d**) plot of the difference between pairwise estimates of recombination frequency and the multi-point estimate of the recombination frequency on the LG25 map plotted against the logarithm of the odds (LOD) score associated with that estimate; (**e**) comparative view of the position of markers on the map and recombination frequency estimates to other markers, with lower to higher recombination frequency (*r*) on a scale of green to red, respectively; (**f**) plot similar to e) showing the LOD values for the markers with a scale of lower to higher LOD from green to red respectively; (**g**) integrated linkage maps of all the 29 linkage groups with the different classes of markers highlighted in different colors. The degree of stress on a particular location on a linkage map is shown by minor green bars adjacent to the position on each linkage group and the quantitative value of the stress is indicated in the figure.

**Figure 3 pathogens-09-00967-f003:**
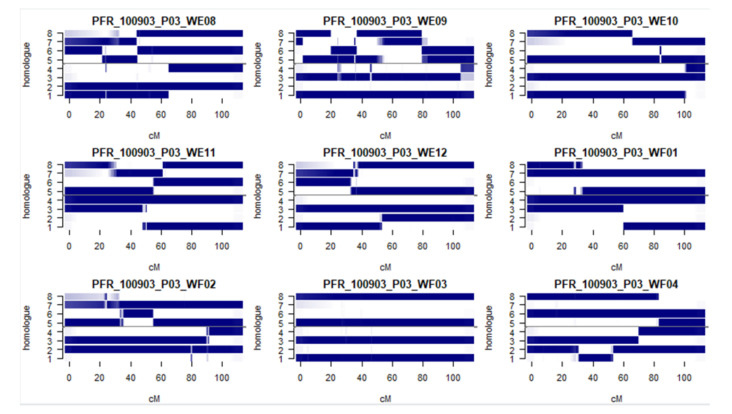
Identity-by-descent (IBD) probabilities for Linkage Group 1 for nine seedlings from the tetraploid kiwifruit population. The figure shows IBD haplotypes in Linkage Group1 for nine F1 individuals in the P1 × P2 tetraploid *A. chinensis* population. The dark blue haplotypes indicate regions of high confidence (probabilities close to or equal to 1).

**Figure 4 pathogens-09-00967-f004:**
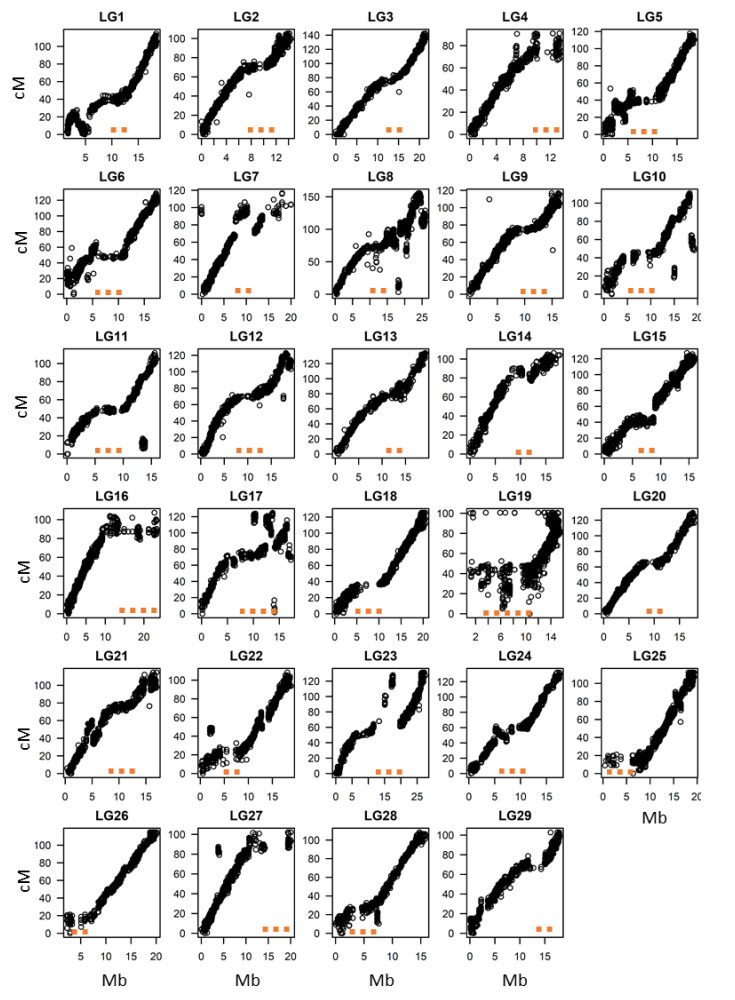
Marey plots for individual linkage groups of tetraploid *A. chinensis*. The figure represents a graphical overview of the genetic map (cM) position on the y-axis when plotted versus the physical map position (Mb) on the x-axis, using the positional information of each marker. The orange dots represent regions where the genetic map is not linear with the physical map, indicating potential regions of reduced recombination frequency (putative regions of pericentromeric heterochromatin).

**Figure 5 pathogens-09-00967-f005:**
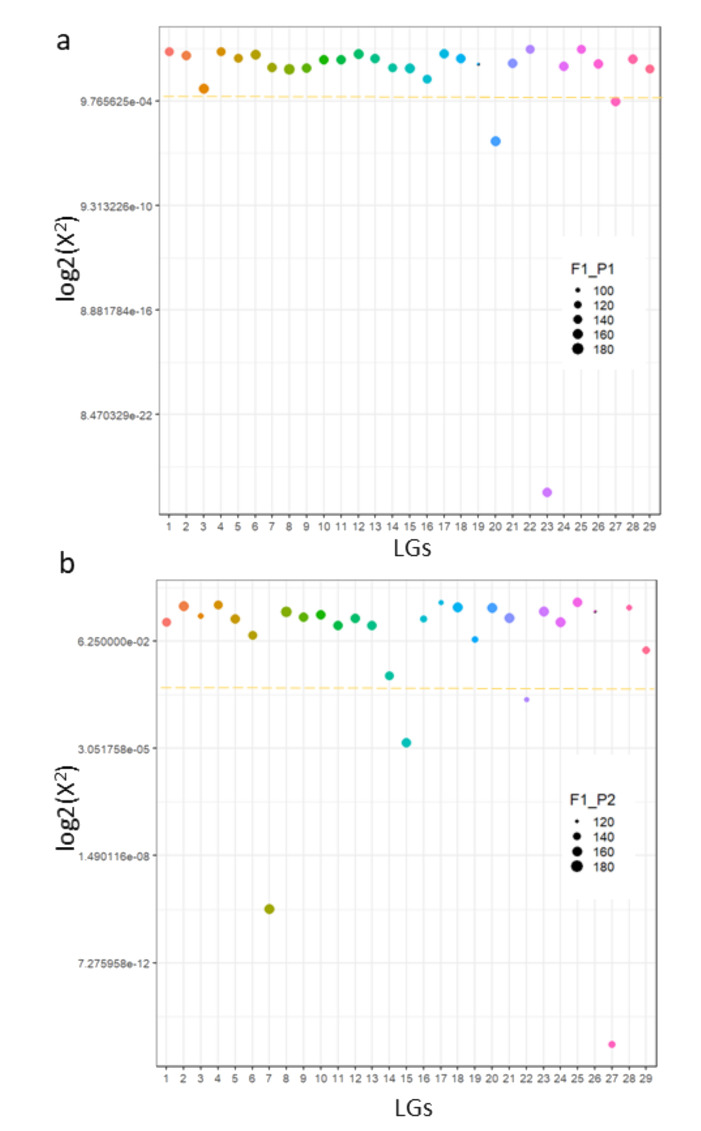
Evidence for polysomic inheritance in tetraploid *A. chinensis*. Figures (**a**,**b**) show ranking of linkage groups (LGs) for pairing in Parent 1 and 2, respectively. Pairing is calculated by X^2^ values generated by polyqtlR for each LG in each parent, with an arbitrary threshold of X^2^ < 0.001 set as indicative of preferential pairing. This is represented by the yellow dotted line and LGs above it are considered to show polysomic inheritance whereas LGs below it are likely to be disomic. The log_2_ (X^2^) is plotted on the Y-axis with LGs on the x-axis differentiated by color. The size of points in the scatter plot is determined by the number of individuals from the population that were used for each parent and referred to as F1-P1 or F1_P2. These individuals showed an unambiguous pairing configuration.

**Figure 6 pathogens-09-00967-f006:**
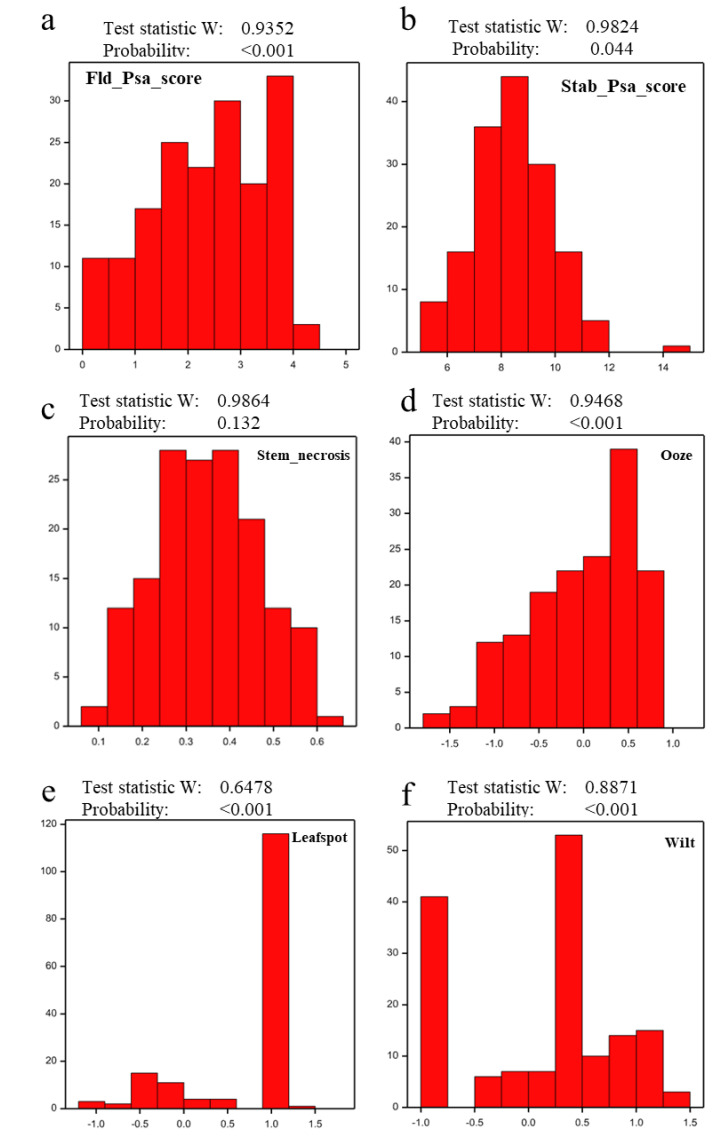
Distribution of phenotypes of plants in the ‘P1 × P2’ population of 188 seedlings of tetraploid *A. chinensis* in response to natural *Pseudomonas syringae* pv. *actinidiae* (Psa) infection and to the stab assay. The x-axis displays the progression of susceptibility from left to right, while the y-axis represents counts for each value of the trait on the x-axis. (**a**) Best linear unbiased predictor (BLUP) values for progressive scores of Psa symptoms in genotypes recorded in the orchard over two years in response to natural Psa infection (Fld_Psa_score); (**b**–**f**) represent Least squares mean (LSM) of scores in the stab assay (Stab_Psa_score, Stem_necrosis, Ooze, Leafspot and Wilt, respectively). The test statistic W is from the Shapiro-Wilks test for the null hypothesis that the distribution is Normal. Phenotypic scores with *p*  <  0.001 do not exhibit Normal distribution.

**Figure 7 pathogens-09-00967-f007:**
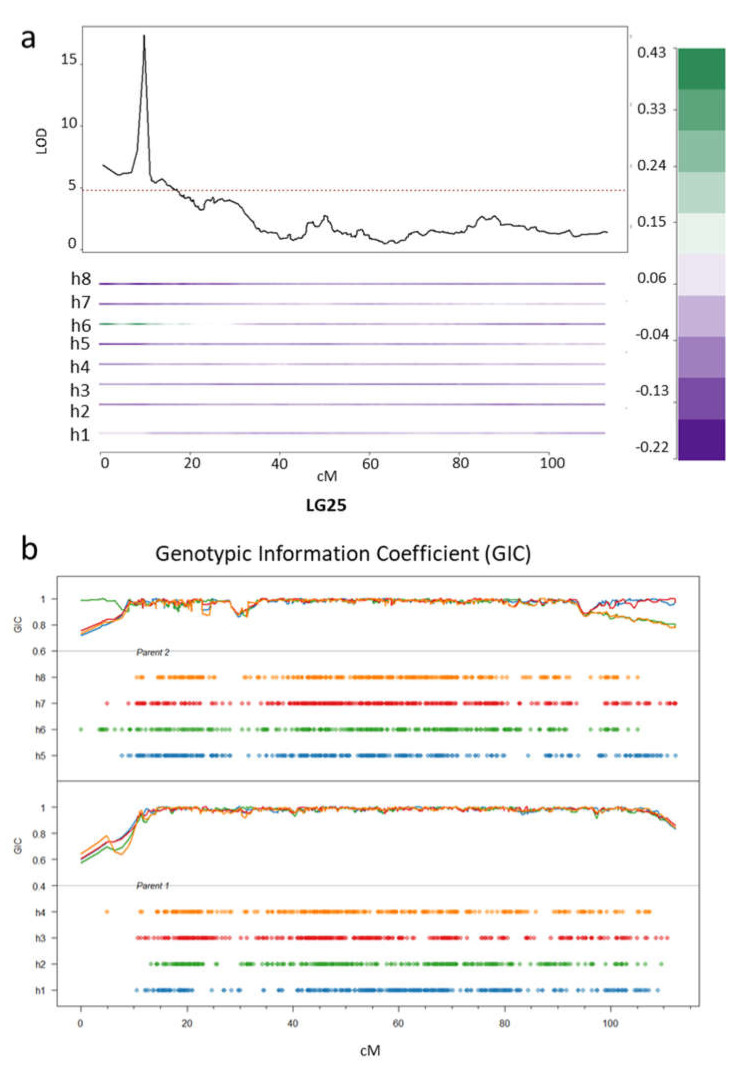
Quantitative Trait Locus (QTL) mapping of the sex of the flowers. (**a**) a QTL for the sex phenotype in the tetraploid *A. chinensis* population on LG25. The red dotted line on the QTL profile indicates the logarithm of the odds (LOD) threshold at α = 0.05. Below the QTL profile is information on the source of the allele from the parental homologues. P1 homologues are represented as h1–h4 and P2 homologues as h5–h8. The green color represents the positive contribution of the allele h6 from P2 for the sex phenotype; (**b**) the profile of the genotypic-information coefficient (GIC) for each parent, in two separate panels. Each line with different colored dots represents the coverage of the markers across the homologues in an integrated genetic map for each parent.

**Figure 8 pathogens-09-00967-f008:**
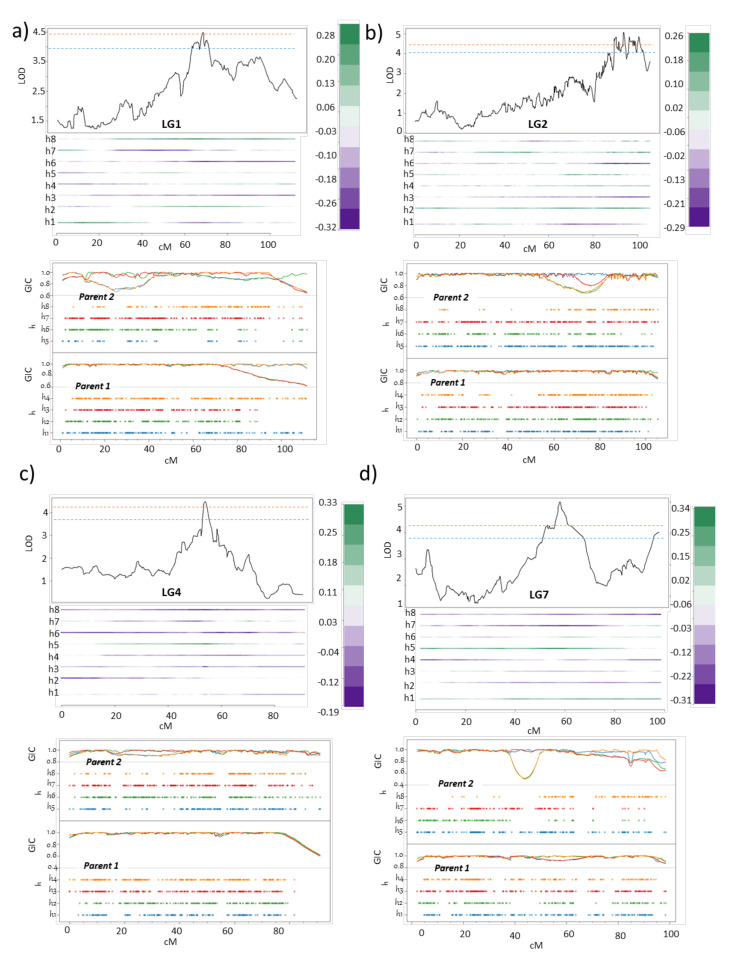
Major quantitative trait loci (QTLs) for control of *Pseudomonas syringae* pv. *actinidiae* (Psa) resistance in the field. QTL and haplotype profile for four QTLs for Fld_Psa_score on (**a**) linkage group (LG)1, (**b**) LG2, (**c**) LG4 and (**d**) LG7, with corresponding profiles of the genotypic-information coefficient (GIC). The orange and blue dashed line indicates the chromosome-wide logarithm of the odds (LOD) threshold at α = 0.05 and 0.1, respectively. Just below the QTL profile is information on the location on the genetic map of the alleles contributing to the phenotype, as well as the source of the alleles from the parental homologues. P1 homologues are represented as h1–h4 and P2 homologues as h5–h8. The positive score/effect of the alleles from the respective homologue is identified by the strong green color, and the quantitative measure for this effect is provided in the legend beside the QTL graph. In addition to QTL information, the graphs are supplemented with the GIC profile of each parent, in two separate panels for each LG.

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
