# Peer review of "QTL Mapping for Resistance to Cankers Induced by Pseudomonas syringae pv. actinidiae (Psa) in a Tetraploid Actinidia chinensis Kiwifruit Population"

_pathogens, 2020, doi:10.3390/pathogens9110967_

Round 1

Reviewer 1 Report

The authors describe their detailed genetic analysis of a tetraploid Actinidia chinensis population to map resistance to Psa. They use capture sequencing to target the exome allowing them to construct ultra-dense genetic maps in a tetraploid kiwifruit population. They use identity-by-descent trait QTL mapping to relate the contribution of specific alleles from homologues in the male parent, to the control of Psa resistance in the progeny. Finally, they identify genes underlying their identified QTL that allow them to suggest a mechanism for Psa resistance. 

This manuscript is well written and the analysis is well-conducted. This is a particularly difficult problem to tackle (and to explain in text) due to the “segmental” allopolyploid nature of the tetraploid kiwifruit. As such, I only have minor comments but do feel that the manuscript would benefit from some clarifications to ease readability mainly for those who are not expert, particularly when dealing with such genome complexity:

-Particularly in the section 2.3. Linkage Map construction , where the authors talk in detail about segregation types e.g. 1x0, 2x0, 0x1, 1x1, 2x1, 0x2, 1x2, 2x2, 1x3 and alike, this could be difficult for non experts to understand without a bit more background, perhaps a schematic to show how these segregation types are formed/what they represent-or clear reference to articles that explain this would help. 

-The authors list the QTLs identified and the genes underlying these QTLs in the text; additionally it would be very useful to have this information in a Table for ease of look up for the reader

Reviewer 2 Report

This is a detailed investigation of disease resistance in a tetraploid kiwifruit population.  The methods will be useful for other researchers working on polyploid plant species.  Most of the comments in the attached document are suggestions to improve the clarity of presentation.  
